# Bacterial community structure of *Anopheles hyrcanus* group, *Anopheles nivipes*, *Anopheles philippinensis*, and *Anopheles vagus* from a malaria-endemic area in Thailand

Patcharaporn Boonroumkaew[1,2], Rutchanee Rodpai[1,2], Atiporn Saeung[3], Kittipat Aupalee[3], Jassada Saingamsook[3], Petchaboon Poolphol[4], Lakkhana Sadaow[1,2], Oranuch Sanpool[1,2], Penchom Janwan[5], Tongjit Thanchomnang[6], Pewpan M. Intapan[1,2], Wanchai Maleewong[1,2]*

1 Department of Parasitology, Faculty of Medicine, Khon Kaen University, Khon Kaen, Thailand, 2 Mekong Health Science Research Institute, Khon Kaen University, Khon Kaen, Thailand, 3 Center of Insect Vector Study, Department of Parasitology, Faculty of Medicine, Chiang Mai University, Chiang Mai, Thailand, 4 The Office of Disease Prevention and Control Region 10th, Ubon Ratchathani, Thailand, 5 Department of Medical Technology, School of Allied Health Sciences, Walailak University, Nakhon Si Thammarat, Thailand, 6 Faculty of Medicine, Mahasarakham University, Kham Riang, Maha Sarakham, Thailand

* wanch_ma@kku.ac.th

**Data Availability Statement:** All sequence reads have been deposited at the NCBI Sequence Read

## Abstract

Bacterial content of mosquitoes has given rise to the development of innovative tools that influence and seek to control malaria transmission. This study identified the bacterial microbiota in field-collected female adults of the *Anopheles hyrcanus* group and three *Anopheles* species, *Anopheles nivipes*, *Anopheles philippinensis*, and *Anopheles vagus*, from an endemic area in the southeastern part of Ubon Ratchathani Province, northeastern Thailand, near the Lao PDR-Cambodia-Thailand border. A total of 17 DNA libraries were generated from pooled female *Anopheles* abdomen samples (10 abdomens/ sample). The mosquito microbiota was characterized through the analysis of DNA sequences from the V3−V4 regions of the 16S rRNA gene, and data were analyzed in QIIME2. A total of 3,442 bacterial ASVs were obtained, revealing differences in the microbiota both within the same species/group and between different species/group. Statistical difference in alpha diversity was observed between *An. hyrcanus* group and *An. vagus* and between *An. nivipes* and *An. vagus*, and beta diversity analyses showed that the bacterial community of *An. vagus* was the most dissimilar from other species. The most abundant bacteria belonged to the Proteobacteria phylum (48%-75%) in which *Pseudomonas*, *Serratia*, and *Pantoea* were predominant genera among four *Anopheles* species/group. However, the most significantly abundant genus observed in each *Anopheles* species/group was as follows: *Staphylococcus* in the *An. hyrcanus* group, *Pantoea* in the *An. nivipes*, *Rosenbergiella* in *An. philippinensis*, and *Pseudomonas* in *An. vagus*. Particularly, *Pseudomonas* sp. was highly abundant in all *Anopheles* species except *An. nivipes*. The present study provides the first study on the microbiota of four potential malaria vectors as a starting step towards understanding the role of the microbiota on mosquito biology and ultimately the development of potential tools for malaria control.

Archive (SRA) under project accession number PRJNA953178.

**Funding:** This research was supported by the Fundamental Fund of Khon Kaen University from the National Science, Research and Innovation Fund (NSRF). The funders had no role in study design, data collection and analysis, decision to publish, or preparation of the manuscript.

**Competing interests:** The authors have declared that no competing interests exist.

## Introduction

Several *Anopheles* species are medically important vectors of certain infectious agents, including protozoan parasites of the genus *Plasmodium* that cause human malaria. Malaria is a widespread and life-threatening disease that results in high levels of morbidity and mortality with an estimated 247 million cases in 2021 from 84 malaria-endemic countries in African, Central and South American, and Asian regions including Thailand [1]. A total of 77 formally named *Anopheles* species have been reported in Thailand [2, 3, 4, 5], but only seven of these (i.e., *Anopheles dirus*, *Anopheles baimaii*, *Anopheles minimus*, *Anopheles aconitus*, *Anopheles maculatus*, *Anopheles sawadwongporni*, and *Anopheles pseudowillmori*) are considered significant malaria vectors [6]. In eastern Thailand, *Anopheles epiroticus* is reported as a secondary vector, while members of the *Anopheles barbirostris* complex are incriminated as potential vectors in the western region of the country [6]. The *Anopheles hyrcanus* group, as well as *Anopheles nivipes*, *Anopheles philippinensis*, and *Anopheles vagus*, are also described as secondary and/or potential vectors of malaria parasites in many Asian countries, including India [7, 8], Bangladesh [9], Myanmar [10], Cambodia [11], China-Laos border regions [11], and Thailand [11, 12]. Furthermore, some *Anopheles* species are regarded as vectors of filarial nematode diseases, including lymphatic filariasis, mansonellosis, and loiasis [13].

Mosquito gut microbiota can influence the development, digestion, immunity, metabolism, and other physiological functions of their hosts [14, 15]. Several studies in African countries, including Kenya [16, 17], Ghana [18], Senegal [19], Mali [20], Ethiopia [21], Burkina Faso [22, 23], Cameroon [24] and the Republic of Guinea [22], as well as in Asian countries, i.e., Vietnam [25], Thailand [26], China [27], and India [28, 29] have shown that variations in gut microbiota affect the ability of insects, especially *Anopheles* mosquitoes, to transmit pathogens.

To accelerate progress towards malaria elimination, the World Health Organization (WHO) has launched the Global Technical Strategy (GTS) for 2030, which aims to decrease global malaria incidence and mortality rates by at least 90% by 2030 [30]. One of these strategies is to control mosquito populations by using biocontrol tools that involve the natural microbial communities associated with mosquitoes [31], which have a possible impact on malaria transmission and severity [32].

The aim of this study is to characterize the bacterial community structure of four *Anopheles* species/group from an endemic area of Thailand based on V3 and V4 regions of the 16S rRNA gene. This study sheds new light on the microbiota of these potential malaria vectors, which is crucial for understanding the role of the microbiota in mosquito biology. Ultimately, the findings of this study can also assist in the development of potential tools for controlling mosquito-borne diseases.

## Materials and methods

### Ethics statement

The protocol related to human (No. 291/2019) and animal (No. 16/2019) used in this study was approved by the Research Ethics Committee of the Faculty of Medicine, Chiang Mai University, Thailand. This study was also approved by the Animal Ethics Committee of the Faculty of Medicine, Khon Kaen University, Thailand (AMEDKKU 012/2022).

### Female adult mosquito collections and species identification

Between 2019 and 2021, adult mosquitoes were collected using two methods: human landing catches (HLC) and buffalo bait collections (BBC) at Ban Huai Kha village, Buntharik District, Ubon Ratchathani Province (14.6029N, 105.3893E) (Fig 1) on two consecutive nights. The mosquitoes were collected throughout the study period, covering three seasons (hot, rainy, and cold). Briefly, HLC was performed only outdoors close to homes. One team of two human collectors gathered mosquitoes between the hours of 18:00 and 24:00, and a second team did this between 00:00 and 06:00. For buffalo bait collection (BBC), one buffalo was tethered and surrounded by a bed net suspended 30 cm above the ground level. The buffalo was exposed to mosquitoes entering the net uninterrupted for 45 minutes each hour [12]. All *Anopheles* mosquitoes resting on inside walls of the net after having bitten the buffalo were collected using an aspirator during the remaining 15 minutes each hour. The ambient air temperature and relative humidity were recorded each hour of collection using a digital hygro-thermometer. All *Anopheles* mosquitoes were morphologically identified to species group, complex, or species under stereomicroscopes using illustrated morphological keys [2]. It is very difficult to identify species within the *An. hyrcanus* group based on morphology alone [33]. Then, we termed our specimens belonging to *An. hyrcanus* group. The head and thorax were separated from the abdomen using sterile forceps for detection of *Plasmodium* parasites by molecular method [34]. The four most abundant species/group collected, including *An. hyrcanus* group, *An. nivipes*, *An. philippinensis*, and *An. vagus* were selected for the following experiments. Ten abdomens of each *Anopheles* species collected on the same day at around the same time were pooled per sample; *An. hyrcanus* group (HYR; $n$ = 4 pools), *An. nivipes* (NIV; $n$ = 5 pools), *An. philippinensis* (PHI; $n$ = 4 pools), and *An. vagus* (VAG; $n$ = 4 pools) and kept at -20°C until DNA extraction for further investigation of malaria and bacterial community structure.

### DNA extraction from abdomens and detection of *Plasmodium* spp.

Genomic DNA extractions were performed using the PureLink® Genomic DNA Mini Kit (Invitrogen, Carlsbad, CA, USA) according to the manufacturer's instructions. DNA concentration and purity were monitored using a NanoDrop Spectrophotometer (Thermo Fisher Scientific, Waltham, MA, USA).

The presence of *Plasmodium* spp. DNA was investigated based on partial cox1 gene amplification using DNA samples extracted from separated abdomen specimens. The PCR reactions were performed following Echeverry et al. (2017) [34] with some modifications. Briefly, each PCR reaction was carried out in a 25 μL reaction volume, containing: 1 unit of Platinum Taq DNA polymerase (Invitrogen, Carlsbad, CA, USA), 2 μL of 2.5 mM dNTPs mix (200 μM) (New England Biolabs, Ipswich, Massachusetts, USA), 0.75 μL of 50 mM MgCl$_2$ (1.5 mM), 2.5 μL of 10x PCR buffer (1x) (Invitrogen), 1 μL of 10 μM each of COX-IF (5′ -AGA ACG AAC GCT TTT AAC GCC TG-3′) (0.5 μM) and COX-IR (3′-ACT TAA TGG TGG ATA TAA AGT CCA TCC wGT-5′) primers (0.5 μM), 1 μL of DNA sample, and made up to 25 μL with sterile water. The amplification process commenced at 94°C with a 5 min heat-activation step, followed by 40 cycles of 94°C for 1 min, 62°C for 1 min and 72°C for 90 s with a 10 min final extension step at 72°C. Along with every PCR performed, DNA extracted from laboratory *Plasmodium falciparum* and *Plasmodium vivax* in-vitro cultures was used as positive controls.

### Bacterial 16S rRNA gene amplification and sequencing

Seventeen DNA samples from pools of ten abdomens of each *Anopheles* species: *An. hyrcanus* group (HYR; $n$ = 4 pools), *An. nivipes* (NIV; $n$ = 5 pools), *An. philippinensis* (PHI; $n$ = 4 pools),

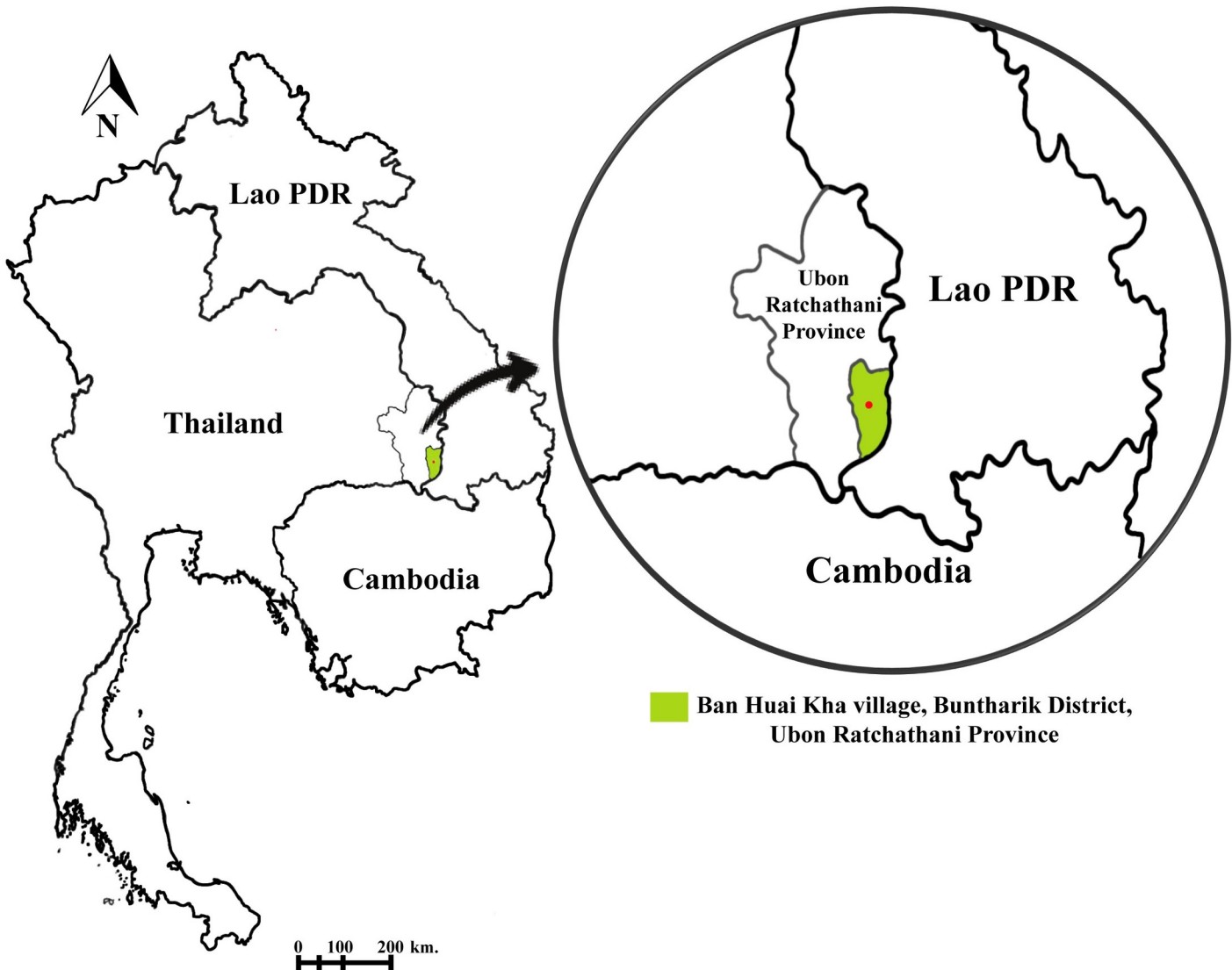

**Fig 1. Location of study site at Ban Huai Kha village, Buntharik District, Ubon Ratchathani Province, Thailand.**

Ban Huai Kha village, Buntharik District, Ubon Ratchathani Province

and *An. vagus* (VAG; $n$ = 4 pools) were used for amplification and sequencing. The universal region-specific primers 341F (5'-CCT AYG GGR BGC ASC AG-3') and 806R (5'-GGA CTA CNN GGG TAT CTA AT-3') (NovogeneAIT Genomics Pte. Ltd., Singapore) tagged with sample-identifying barcodes to amplify the V3–V4 regions of the 16S rRNA gene were used. PCR products of approximately 400–450 bp in length were generated. PCR products of the expected size were selected by 2% agarose gel electrophoresis and used for library preparation. At each step of the process, quality control was carried out to maintain the accuracy and reliability of the sequencing data according to the Illumina sequencing system (NovogeneAIT Genomics). The libraries were generated by end-repair, A-tailing, and ligation with Illumina adapters, then assessed using Qubit and qPCR for quantification and a bioanalyzer for size distribution detection. Quantified libraries were sequenced on an Illumina paired-end platform (performed by NovogeneAIT Genomics Pte. Ltd., Singapore) to generate 250 bp paired-end raw reads.

## Bioinformatics and statistical analyses

The 16S rRNA gene reads were processed using the QIIME2 pipeline (https://qiime2.org/). The paired-end reads were assigned to samples based on their unique barcodes, truncated by cutting off the barcode and primer sequences, and merged using FLASH (version 1.2.11, http://ccb.jhu.edu/software/FLASH/) [35]. This process generated the raw reads. Quality filtering of these was performed using the fastp software (version 0.20.0) to obtain high-quality clean reads [36]. The reads were compared with the Silva database (https://www.arbsilva.de/) using Vsearch (version 2.15.0) [37] to detect chimeric sequences, which were removed to obtain the effective reads used for subsequent analysis [38]. The sequences have been deposited in the NCBI database under the accession number PRJNA953178.

The effective reads were denoised using DADA2 in the QIIME2 program (version 2020.06) to obtain initial amplicon sequence variants (ASVs). Note that a single species can have more than one ASV associated with it if there is intra-specific variation in this portion of the 16S rRNA gene. Each ASV was compared with the Silva database using the classify-sklearn algorithm in QIIME2 software to obtain the taxonomy annotation [39, 40] and the abundance of each taxon at the level of kingdom, phylum, class, order, family, genus, and species. Moreover, the abundance of ASVs was used to generate a rarefaction curve for estimating the species/ group richness and diversity in the microbiota of four *Anopheles* species/group. Alpha diversity within and between groups was calculated in QIIME2, including observed-species, Chao1, Shannon, Simpson, ACE, and Faith's alpha metrics. Alpha diversity comparisons were performed using the Kruskal–Wallis nonparametric test. Beta diversity was investigated by weighted UniFrac distance and unweighted UniFrac distance to measure the differences between samples (implemented in QIIME2). The results displayed graphically here were visualized using R software (version 3.5.3).

## Results

### Overall distribution of bacteria within different *Anopheles* species

A total of 2,659 *Anopheles* females belonging to one group (*An. hyrcanus* group) in the subgenus *Anopheles* and nine species/group (*An. dirus* s.l., *An. karwari*, *An. kochi*, *An. maculatus* group, *An. minimus* s.l., *An. nivipes*, *An. philippinensis*, *An. tessellatus* and *An. vagus*) in the subgenus *Cellia* were collected throughout study period. The four most abundant taxa were *An. hyrcanus* group (78.15%) followed by *An. philippinensis* (14.70%), *An. nivipes* (5.00%) and *An. vagus* (0.90%). No *Plasmodium* DNA was discovered using PCR in any of the 17 pooled female *Anopheles* abdomen samples. Bacterial diversity and abundance were calculated separately for each pool: *An. hyrcanus* group (HYR; $n$ = 4 pools), *An. nivipes* (NIV; $n$ = 5 pools), *An. philippinensis* (PHI; $n$ = 4 pools), and *An. vagus* (VAG; $n$ = 4 pools). The output statistics are shown in S1 Table. A total of 2,033,707 effective reads (S1 Table) was obtained from all samples, with the number of reads per pool ranging from 74,854 to 143,839. They were assigned to 3,442 bacterial ASVs (S2 Table). The alpha rarefaction curve represents the sequence to a sufficient depth of the sample, with most reaching saturation, which was sufficient to capture the scope of microbial diversity and indirectly reflect the abundances of our *Anopheles* species/group (S1 Fig). Overall, the microbiota of four *Anopheles* species/group consisted of taxa belonging to 36 phyla, 90 classes, 223 orders, 368 families, 747 genera, and 598 species (S3 Table). The greatest number of ASVs was found in *An. vagus*. Twenty-eight ASVs were present in all samples, and each *Anopheles* species also had some unique ASVs (Fig 2 and S4 Table). Shared ASVs were dominant bacteria and a small number of ASVs were different between samples in the same *Anopheles* species/group (S5 Table).

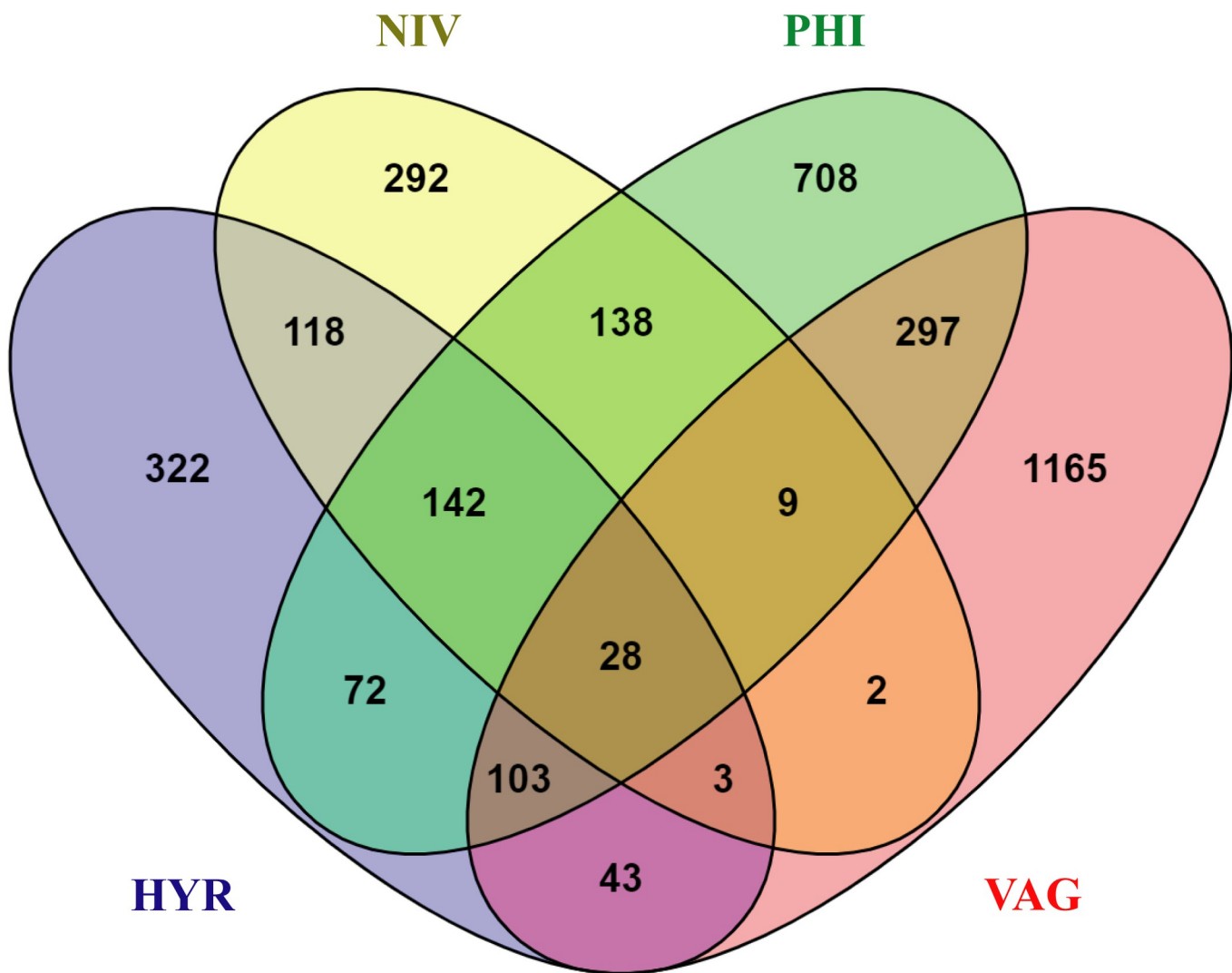

**Fig 2. Venn diagram plot indicates unique and shared amplicon sequence variants (ASVs) among the four *Anopheles* species/group.** Each ellipse represents one group. Values represent the number of ASVs in each overlapping or unique segment.

## Bacterial microbiota composition within different *Anopheles* species

The microbiota of four *Anopheles* species/group were differed between the same and between species/group (S4 and S5 Tables). Proteobacteria were found in most samples as the main constituent of a shared and conserved core microbiota at the phylum level, accounting for 47.5% of reads in *An. hyrcanus* group, 60.0% in *An. nivipes*, 75.4% in *An. philippinensis*, and 67.6% in *An. vagus*. Firmicutes were the second most abundant phylum, with the other seven phyla ranging from 9.8% to less than 1% in abundance (Fig 3A).

At the family level, the microbiota revealed more variation between the different *Anopheles* species. The Pseudomonadaceae was dominant in *An. vagas* and *An. hyrcanus* group, whereas Erwiniaceae was dominant in *An. philippinensis* and *An. nivipes*, followed by Moraxellaceae. Carnobacteriaceae and Burkholderiaceae were abundant in *An. hyrcanus* group and *An. vagas*, respectively (Fig 3B).

Seven of the ten most abundant genera of the microbiota were gram-negative taxa, including *Pseudomonas*, *Serratia*, *Rosenbergiella*, *Ralstonia*, *Acinetobacter* and *Pantoea*. Gram-positive

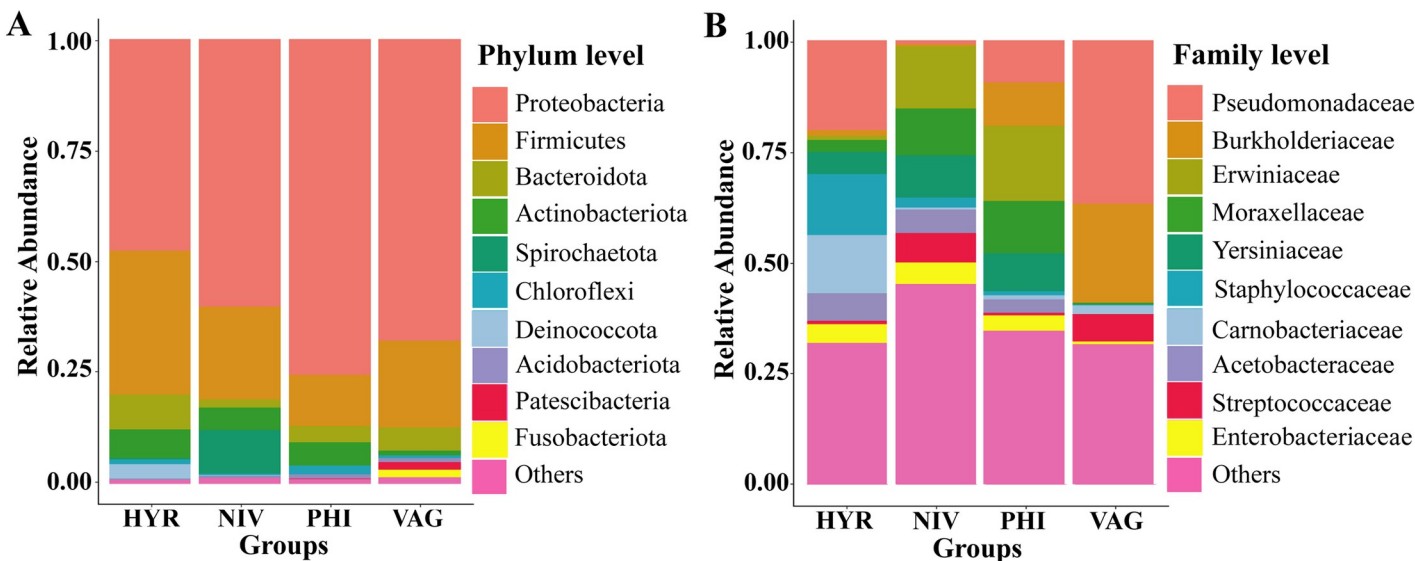

**Fig 3. Composition of bacterial communities of four *Anopheles* species/group.** The top ten taxa in terms of relative abundance at phylum level (**A**) and family level (**B**).

bacteria were represented by the genera *Staphylococcus* and *Carnobacterium*. The most abundant genera represented in *An. hyrcanus* group included *Elizabethkingia*, *Staphylococcus*, *Glutamicibacter*, *Carnobacterium*, *Thermus*, and *Asaia*. For *An. nivipes*, the most abundant genera included *Cutibacterium*, *Lysinibacillus*, *Pantoea*, *Lactococcus*, and *Novispirillum*. For *An. philippinensis*, *Rosenbergiella* was the highest significant, followed by *Bacillus*, *Acinetobacter*, and *Romboutsia*. For *An. vagus*, the most abundant genera included *Pseudomonas*, *Porphyromonas*, *Veillonella*, *Granulicatella*, *Streptococcus*, *Massilia*, *Ralstonia*, and *Cupriavidus* (Fig 4A).

*Pseudomonas* sp. was very abundant in all mosquito species except *An. nivipes*. *Carnobacterium maltaromaticum* and *Pseudomonas aeruginosa* were found to be abundant in *An. hyrcanus* group. The most abundant species found in *An. nivipes* was *Spironema culicis*, followed by *Lactococcus lactis* and *Novispirillum itersonii*. *Elizabethkingia meningoseptica*, *Asaia krungthepensis*, *Cedecea neteri*, and *Staphylococcus saprophyticus* were abundant in all species except *An. nivipes* (Fig 4B).

## Bacterial community richness and diversity

For alpha diversity, observed species, ACE, and Chao1 indices that measure bacterial community richness, and the phylogenetic diversity (Faith's PD) index all revealed a significant difference between *An. hyrcanus* group and *An. vagus* and between *An. nivipes* and *An. vagus* ($p < 0.05$ in each case). In addition, the bacterial diversity was assessed using the Shannon and Simpson indices, which showed no significant differences between the *Anopheles* species/ group (Fig 5).

The differences of bacterial communities between samples were analyzed in terms of beta-diversity. The dissimilarity coefficient between pairwise sample groups was plotted using unweighted and weighted UniFrac distance matrix which showed beta diversity ranging from 0.51 to 0.75 (Fig 6A) and from 0.24 to 0.43 (Fig 6B), respectively.

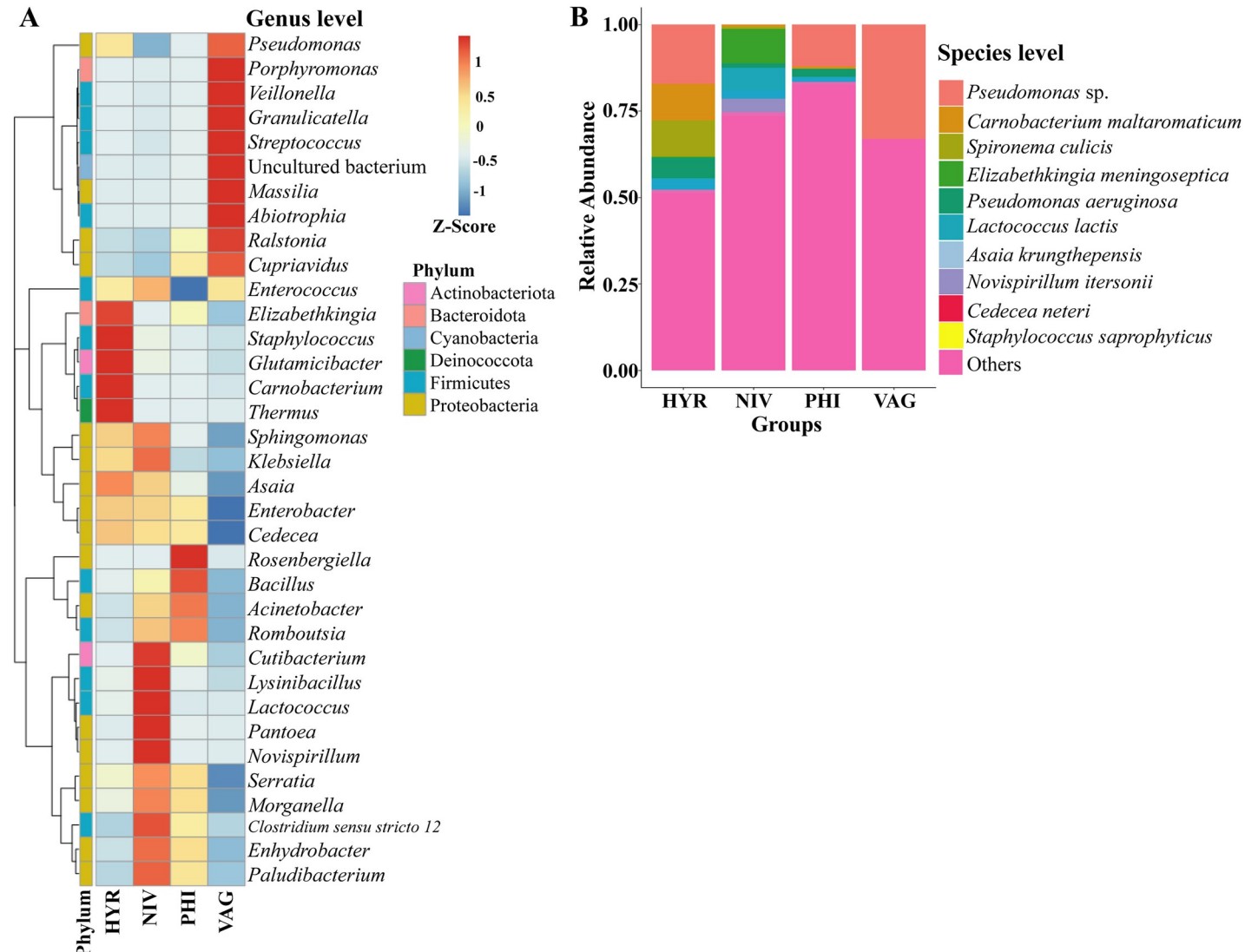

**Fig 4.** Taxonomic abundance cluster heatmap showing the relative abundance of the 35 most abundant genera (A). The ten most abundant taxa at species level (B).

## Discussion

Field-collected female adults of four *Anopheles* species/group were sampled from a malaria-endemic area in the southeastern part of Ubon Ratchathani Province, northeastern Thailand, near the Lao PDR-Cambodia-Thailand border [41]. All samples were negative for *Plasmodium* DNA by PCR. For this result, we were not able to analyze and compare the bacterial microbiota between non-infected and infected *Anopheles* mosquitoes. However, the limit information on the gut microbiota of these four *Anopheles* species/group were obtained, which was required for further study when positive malaria samples are available for comparison. There were several studies that investigated the correlations between midgut microbiota and the mosquito malaria infection status [23, 42]. The abundance of Enterobacteriaceae in the midgut of *An. gambiae* correlated significantly with *P. falciparum* infection status [42]. A positive correlation between the Weeksellaceae and Acetobacteraceae families and the presence of *Plasmodium* gametocytes in the blood meal was also observed in *An. gambiae* s.l. [23]. Additionally,

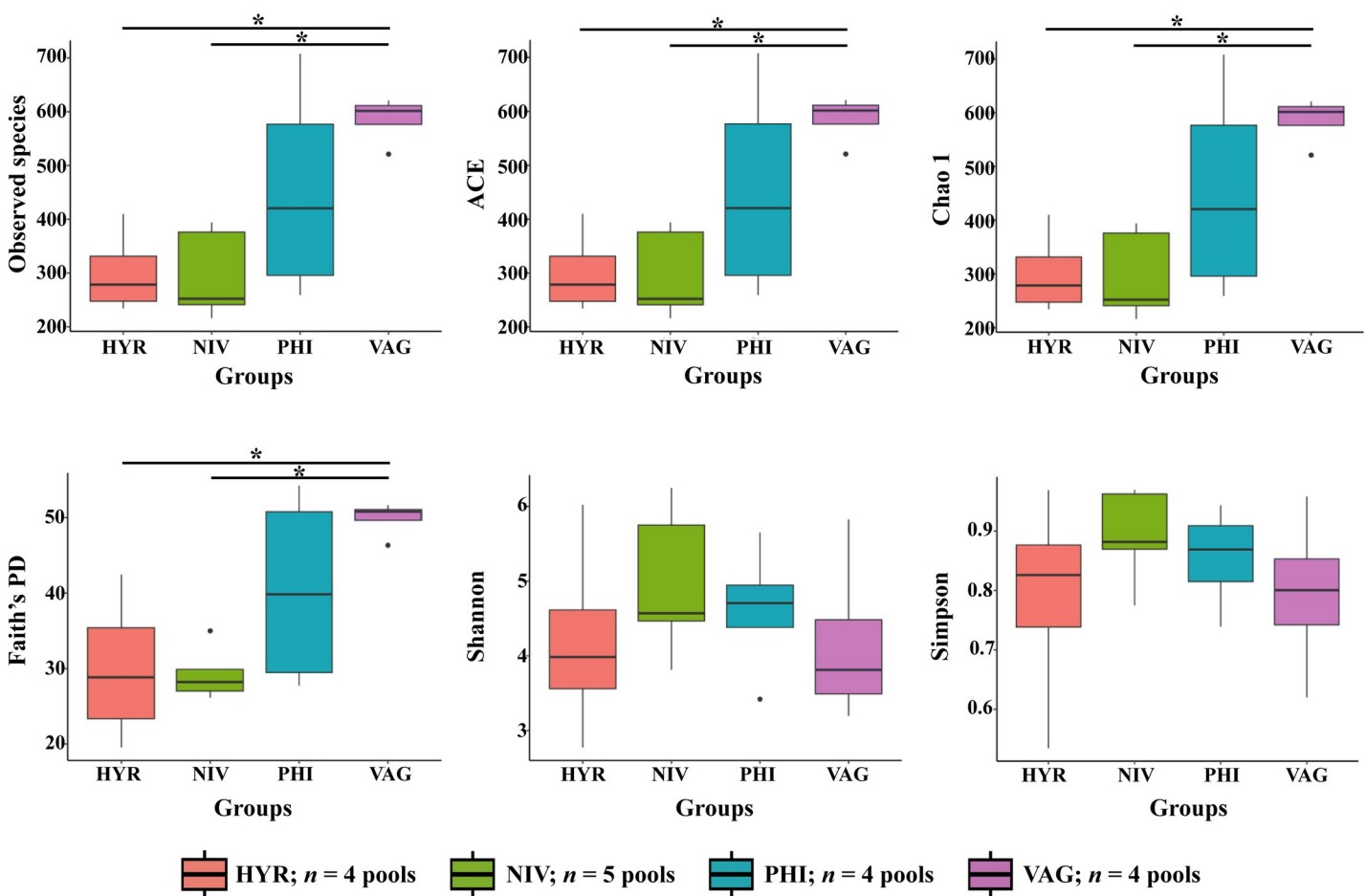

**Fig 5. Boxplot of alpha diversity indices.** Observed species; the number of species directly observed; the higher the index, the more species detected, ACE and Chao1; the indices estimate the species richness in sample groups, Faith's PD; a phylogenetic diversity index, Shannon and Simpson; the indices reflect the ASVs diversity within and among sample pools. Kruskal-Wallis-Pairwise were used to detect statistically significant differences between mosquito species (*indicates $p$ value < 0.05).

five bacterial genera, including *Aerococcus*, *Megasphaera*, *Peptostreptococcus*, *Roseomonas* and *Streptococcus* were detected exclusively in the abdomen of *P. vivax*-infected *An. minimus* [26].

Here, the abdominal bacterial community structure of four *Anopheles* species/group was profiled using the 16S rRNA gene V3–V4 regions. The relative abundance of the main bacterial genera varied among *Anopheles* species. To our knowledge, this is the first report of bacterial diversity in these four *Anopheles* species/group. The bacterial communities differed significantly among the four species/group in terms of alpha diversity and taxonomic diversity at different taxonomic levels. Importantly, the microbiota of *An. vagus* was found to be significantly more diverse than that of the *An. hyrcanus* group and *An. nivipes* in terms of observed species, ACE, Chao1, and phylogenetic diversity (Faith's PD). According to UniFrac distance, *An. vagus* was the most dissimilar from other species. This result possibly be due to the differences microbial diversity in habitats and breeding places between the four *Anopheles* species/group. *Anopheles vagus* larvae are commonly found in a wide variety of groundwater habitats, in water jars and in holes. While the main species in the *An. hyrcanus* group immature stages are likely found in rice fields, marshy and swampy areas, ponds, and other similar habitats that contain emergent vegetation as well as the larvae of *An. nivipes*, and *An. philippinensis* are found in clean water with considerable vegetation [2].

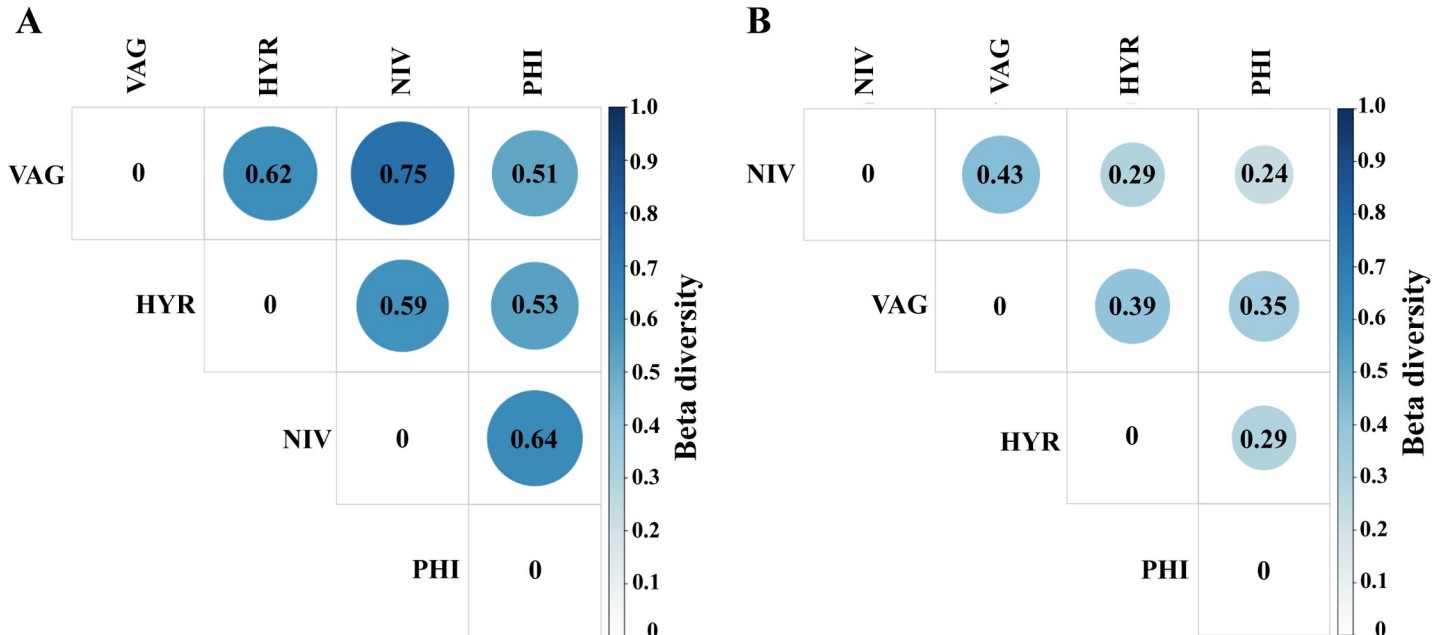

**Fig 6.** Heatmaps of beta diversity matrix of the microbiota were plotted based on unweighted (A) and weighted (B) UniFrac distance matrix to reflect the dissimilarity mosquito species. The smaller the value, the lower the differences in species diversity between the two sample groups.

Proteobacteria and Firmicutes phyla predominated in the midguts of our field-collected *Anopheles*, in agreement with previous studies [25, 42, 43, 44]. The Pseudomonadaceae, a member of the Proteobacteria phylum, was present in all four mosquito species/group but was least abundant in *An. nivipes*. Previous study has reported that blood-fed mosquitoes favor the rapid proliferation of members of the Pseudomonadaceae [45]. It is possible that our field-collected *Anopheles* acquired different blood meals under natural conditions, which correspond to the observed differences in their gut microbiota. Several families (Pseudomonadaceae, Aeromonadaceae and Enterobacteriaceae) reported in this study are core microbiota of many *Anopheles* species in Africa, Asia, and America [46].

The most abundant bacterial genera identified in this study, such as *Enterobacter*, *Aeromonas*, *Pantoea*, *Pseudomonas*, *Elizabethkingia*, *Klebsiella* and *Serratia* were also the most abundant according to previous reports [26, 46]. Among these, some taxa have been suggested as promising candidates for paratransgenic modifications in vector-control strategies [47]. *Serratia* affects *Plasmodium* development in *Anopheles* species, thereby rendering it a potential candidate for the development of a malaria transmission intervention strategy [48, 49, 50, 51, 52]. Similarly, the popular candidate bacteria *Pantoea*, *Enterobacter*, and *Pseudomonas* were able to significantly inhibit the development of *Plasmodium* in the mosquito host [48, 53, 54]. However, there is a notably lower abundance of these bacteria in *An. vagus* compared to the other three *Anopheles* species. These findings indicate that bacterial composition varies among different *Anopheles* species. *Anopheles vagus* was dominated by *Pseudomonas*, *Ralstonia*, *Cupriavidus*, and *Streptococcus*. Previously, *Streptococcus* had been reported as one of five bacterial genera detected in *Anopheles minimus* infected with *Plasmodium vivax* in a malaria-endemic region of western Thailand [26]. The genus *Rosenbergiella* was the most abundant taxon in *An. philippinensis*. This genus is commonly present in flower nectar and in other insects worldwide, such as *Gastrolina depressa* and *Brithys crini* [55, 56], but this is the new record from *Anopheles* species. The factors that affect the high abundance of *Rosenbergiella* in the abdomens of female *An. philippinensis* need further investigation.

At the species level, *An. nivipes* revealed more evenness in bacterial taxa than did other mosquitoes, with *Spironema culicis* being the most abundant. Although *Spironema culicis* has been previously isolated from *Culex nigripalpus*, a vector of Saint Louis Encephalitis and West Nile viruses in Florida, USA [57], it has not been reported to dominate the gut of *Anopheles* mosquitoes before. Significantly, *Pseudomonas aeruginosa* and *Carnobacterium maltaromaticum* were found to be dominant in the abdomens of female *An. hyrcanus* group. *Carnobacterium maltaromaticum* is related to plant defense detoxification mechanisms and pathways [58]. *Pseudomonas aeruginosa* can form biofilm [59] and inhibit *P. falciparum* sporogonic development as previously reported in *An. stephensi* [54]. The presence of these taxa in high abundance in some mosquito species but not in others warrants future investigation into the underlying biological factors explaining this variation.

Our findings provide a snapshot of the microbiota in four *Anopheles* species/group that are potential vectors of malaria in Thailand. More work is required before using these bacteria in any field applications aimed at limiting the transmission of malaria. The microbiota should be further investigated for possible effects on filarial parasites within the mosquito host. In the future, manipulation of the microbiota may become an important prevention strategy for blocking the transmission of *Plasmodium* and/or filarial worms in vectors as well as for developing new diagnostics, treatments, and prevention methods for these parasites in humans.

A limitation of the study is that our samples were collected in the field, with no attempt made to standardize environmental parameters of the collection sites. The microbiota of the mosquitoes may have been influenced by such local parameters. Further work is needed to include control samples from bacterial environments such as the water from their breeding places, etc. in the collection areas to facilitate identification and subsequent removal of contaminant sequences.

## Supporting information

**S1 Table. Results of the processed sequencing data.**
(XLSX)

**S2 Table. Complete list of bacterial amplicon sequence variants found across all sample groups.**
(XLSX)

**S3 Table. List of each taxonomic rank in the bacteria kingdom (phylum, class, order, family, genus, species) across all sample groups.**
(XLSX)

**S4 Table. List of amplicon sequence variants (ASVs) indicates unique and shared ASVs among the four *Anopheles* species/group.**
(XLSX)

**S5 Table. Shared and unique amplicon sequence variants (ASVs) in each *Anopheles* species/group.**
(XLSX)

**S1 Fig.** Rarefaction curves and depth of the richness and diversity indices included the Chao1 (A-B), Faith's PD (C-D), observed species (E-F), and Shannon (G-H) of the microbial communities. A, C, E, and G represented data from each *Anopheles* group. B, D, F, and H represented data from each sample. (HYR: *An. hyrcanus* group; NIV: *An. nivipes*; PHI: *An. philippinensis*; VAG: *An. vagus*).
(TIF)

## Acknowledgments

We thank Dr. David Blair for the English editing of this manuscript.

## Author Contributions

**Conceptualization:** Patcharaporn Boonroumkaew, Rutchanee Rodpai, Atiporn Saeung, Kittipat Aupalee, Jassada Saingamsook, Petchaboon Poolphol, Lakkhana Sadaow, Oranuch Sanpool, Penchom Janwan, Tongjit Thanchomnang, Pewpan M. Intapan, Wanchai Maleewong.

**Data curation:** Patcharaporn Boonroumkaew, Rutchanee Rodpai, Atiporn Saeung, Kittipat Aupalee, Jassada Saingamsook, Petchaboon Poolphol, Lakkhana Sadaow, Oranuch Sanpool, Penchom Janwan, Tongjit Thanchomnang, Pewpan M. Intapan, Wanchai Maleewong.

**Formal analysis:** Patcharaporn Boonroumkaew, Rutchanee Rodpai, Atiporn Saeung, Kittipat Aupalee, Jassada Saingamsook, Petchaboon Poolphol, Lakkhana Sadaow, Oranuch Sanpool, Penchom Janwan.

**Funding acquisition:** Oranuch Sanpool, Wanchai Maleewong.

**Methodology:** Patcharaporn Boonroumkaew, Rutchanee Rodpai, Atiporn Saeung, Kittipat Aupalee, Jassada Saingamsook, Petchaboon Poolphol, Tongjit Thanchomnang, Pewpan M. Intapan, Wanchai Maleewong.

**Project administration:** Wanchai Maleewong.

**Supervision:** Atiporn Saeung, Pewpan M. Intapan, Wanchai Maleewong.

**Writing – original draft:** Patcharaporn Boonroumkaew, Rutchanee Rodpai, Wanchai Maleewong.

**Writing – review & editing:** Patcharaporn Boonroumkaew, Rutchanee Rodpai, Atiporn Saeung, Kittipat Aupalee, Jassada Saingamsook, Lakkhana Sadaow, Oranuch Sanpool, Penchom Janwan, Tongjit Thanchomnang, Pewpan M. Intapan, Wanchai Maleewong.

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
