## [Decision Letter · Decision Letter 0]

30 May 2023

PONE-D-23-12986Bacterial community structure of Anopheles hyrcanus group, Anopheles nivipes, Anopheles philippinensis, and Anopheles vagus from a malaria-endemic area in ThailandPLOS ONE

Dear Dr. Maleewong, 

Thank you for submitting your manuscript to PLOS ONE. After careful consideration, we feel that it has merit but does not fully meet PLOS ONE’s publication criteria as it currently stands. Therefore, we invite you to submit a revised version of the manuscript that addresses the points raised during the review process.

We look forward to receiving your revised manuscript.

Kind regards,

Shawky M Aboelhadid, PhD

Academic Editor

PLOS ONE

Journal Requirements:

4. We note that Figure 1 in your submission contain map/satellite images which may be copyrighted. All PLOS content is published under the Creative Commons Attribution License (CC BY 4.0), which means that the manuscript, images, and Supporting Information files will be freely available online, and any third party is permitted to access, download, copy, distribute, and use these materials in any way, even commercially, with proper attribution. For these reasons, we cannot publish previously copyrighted maps or satellite images created using proprietary data, such as Google software (Google Maps, Street View, and Earth). For more information, see our copyright guidelines: http://journals.plos.org/plosone/s/licenses-and-copyright.

Additional Editor Comments:

The manuscript needs a revision as recommended by the reviewers

Reviewers' comments:

Reviewer's Responses to Questions

**Comments to the Author**

1. Is the manuscript technically sound, and do the data support the conclusions?

Reviewer #1: Partly

2. Has the statistical analysis been performed appropriately and rigorously? 

Reviewer #1: N/A

3. Have the authors made all data underlying the findings in their manuscript fully available?

Reviewer #1: Yes

4. Is the manuscript presented in an intelligible fashion and written in standard English?

Reviewer #1: Yes

5. Review Comments to the Author

Reviewer #1: The manuscript by Wanchai Maleewong et al titled “Bacterial community structure of Anopheles hyrcanus group, Anopheles nivipes, Anopheles philippinensis, and Anopheles vagus from a malaria-endemic area in Thailand” is very important as it is focused on the investigation of the microbiota of some malaria vectors in Thailand. Despite the efforts put in place to control this deadly disease, it remains a public health mainly because of the resistance of mosquitoes to insecticides and the parasites themselves to drugs. Therefore, the use of microbiome based tools has been shown as an alternative vector control strategy. This work is then of great importance.

However, some comments in order to improve the quality are as follows:

Abstract

In general, the authors need to add in the abstract the sample size and the software used for data analysis. Also, indicate if the DNA was extracted from the whole individual mosquito or the pool of mosquitoes, even the sequencing.

In addition, the authors need to state the results of genetic diversity in terms of Alpha diversity and Beta diversity. Finally, it will be better to explain in one line the difference observed between these groups of bacteria.

Line 31: “The results revealed the microbiota of four Anopheles species differed between pools of the same species and between species”. This result needs to be rephrased since Anopheles hyrcanus is stated here as a group and not a species.

Line 32: Before giving the predominant genus for each Anopheles groups, authors should give the predominant genus in the whole mosquito population of An. groups, even the total numbers of the identified bacterial. What about the predominant phylum? Also, the author should add the percentage to recover the overall abundance of these bacterial.

Line 38: “The present study provides new knowledge on the microbiota of four potential malaria vectors as a starting step towards understanding ….”. In the discussion, it is stated that it is the first study done in this context the authors should rewrite this sentence in the abstract such that it clearly appears that it is the first study and not only new knowledge as they said.

Methods

Line 99: The authors said “It is very difficult to identify species within the An. hyrcanus group based on morphology alone”. What about molecular identification? The authors should check if there is no protocol for that.

Line 100: “For that reason, we used our specimens belonging to this species group as An. hyrcanus group”, this sentence is not comprehensible and cannot be linked to the previous. Rewrite it to let it make sense.

Line 102: The authors said “Female Anopheles mosquitoes were dissected for detection and identification of Plasmodium parasites”. The parts dissected should be cited, are they abdomens, Head and thorax or ovaries…? Also, the method used to identify Plasmodium needs to state.

Line 104: “Ten abdomens of each Anopheles species collected on the same day at around the same time were pooled …. bacterial community structure”. How many abdomens one pool? How many pools have been made for each specie? These must be clear in the sentence. Also, add the estimated days collections.

Line 115: “The presence of Plasmodium spp. parasites was investigated based on partial cox1 gene amplification”. The authors should well state the part of the mosquito used here for Plasmodium detection.

Line 126: “From laboratory Plasmodium falciparum and Plasmodium vivax in-vitro cultures was used as 127 positive controls”: Why not used Plasmodium spp as a positive control?

Line 130: “The universal region-specific primers 341F (5'-CCT AYG GGR BGC ASC AG-3') and 806R (5'-GGA CTA CNN GGG TAT CTA AT-3'…V3�V4 regions of the 16S rRNA gene were used.” Reference needs to be added here.

Line 129: “Bacterial 16S rRNA gene amplification and sequencing”. In this paragraph, the authors should clearly give the details on the numbers of samples used, which parts have been used, which Anopheles species and how many pools?

Line 134-138: The authors need to describe in detail the methodology for sample sequencing. Some previous studies have more describe this part. Also indicated whether the negative and positive controls have been used during the sequencing.

Results

In general, the authors need to indicate which season the mosquitoes have been collected, because some previous study show that the season could have a great impact on microbiota diversity.

Line 163: Before talking about the results on the bacterial composition, authors should first give the results of mosquitoes identification. It has been stated in the methodology that the four species used were the more predominant, what about the about? And which are the frequencies?

Line 164. In this paragraph, the authors should mention the quality control statistic: the number of raw reads obtained, the number obtained after filtration, the rarification…

Line 180: “The microbiota of four Anopheles species differed between pools of the same species and between species”. Here the authors mentioned a difference between the pools of the same species, whereas the title on line 179 only mentions the differences between species. Moreover, in the following sentences, the differences they show are only those between species. What are about the differences between the same species?

Line 187: “The Pseudomonadaceae was dominant in An. vagas and An. hyrcanus group, whereas …Carnobacteriaceae and Burkholderiaceae were abundant in An. hyrcanus group and An. vagas, respectively (Fig 3B). This paragraph should be complete, the authors mentioned the predominance in the 4 groups of mosquitoes but for the abundance, they just referred to An. vegas and An. hyrcanus, what about the 2 others? Also, to what the predominance and abundance here referred?

Line 205: “Elizabethkingia meningoseptica, Asaia krungthepensis, …species except An. nivipes”. Is this statement, are the authors sure that Asaia that they found where specifically krungthepensis? If not, they should better write Asaia sp.

Discussion

Line 238: “However, information on the gut microbiota of these four Anopheles species, which was characterized for further study when positive malaria samples are available for comparison”. This sentence must be rewritten; something is missing.

Line 285: “The genus Rosenbergiella was the most abundant taxon in An. philippinensis”. It is just during the discussion that information is clear while it in the results, it is just noticed that this genus is among the most abundant. Since the idea is discussed, this part should be rephrased in the result such the reader is not surprised at the part discussion.

Line 294: “Significantly, Pseudomonas aeruginosa and Carnobacterium maltaromaticum were found to be dominant in the guts of female An. hyrcanus group”. The authors should continue to use the term abdomen rather than gut, there is a slide difference.

6. PLOS authors have the option to publish the peer review history of their article (what does this mean?). If published, this will include your full peer review and any attached files.

Reviewer #1: No

While revising your submission, please upload your figure files to the Preflight Analysis and Conversion Engine (PACE) digital diagnostic tool, https://pacev2.apexcovantage.com/. PACE helps ensure that figures meet PLOS requirements. To use PACE, you must first register as a user. Registration is free. Then, login and navigate to the UPLOAD tab, where you will find detailed instructions on how to use the tool. If you encounter any issues or have any questions when using PACE, please email PLOS at figures@plos.org. Please note that Supporting Information files do not need this step.<quillbot-extension-portal></quillbot-extension-portal>

---

## [Author Response · Author response to Decision Letter 0]

21 Jun 2023

Point by point response to Journal Requirements, Editor and Reviewer comments

Journal Requirements:

Reply: We have done as suggested. We certified the submitted manuscript meets PLOS ONE's style requirements.

Reply: No permits were required for the described study, which complied with all relevant regulations. All information and materials included in our manuscript are the original.

Reply: We have done as suggested. All sequence reads have been deposited at the NCBI Sequence Read Archive (SRA) under project accession number PRJNA953178.Please see https://dataview.ncbi.nlm.nih.gov/object/PRJNA953178?reviewer=ieo2go5gsc7bs0jmhdog9a9r0v

4. We note that Figure 1 in your submission contain map/satellite images which may be copyrighted. All PLOS content is published under the Creative Commons Attribution License (CC BY 4.0), which means that the manuscript, images, and Supporting Information files will be freely available online, and any third party is permitted to access, download, copy, distribute, and use these materials in any way, even commercially, with proper attribution. For these reasons, we cannot publish previously copyrighted maps or satellite images created using proprietary data, such as Google software (Google Maps, Street View, and Earth). For more information, see our copyright guidelines: http://journals.plos.org/plosone/s/licenses-and-copyright.

The following resources for replacing copyrighted map figures may be helpful: USGS National Map Viewer (public domain): http://viewer.nationalmap.gov/viewer/

The Gateway to Astronaut Photography of Earth (public domain):http://eol.jsc.nasa.gov/sseop/clickmap/

Maps at the CIA (public domain): https://www.cia.gov/library/publications/the-world-factbook/index.html and https://www.cia.gov/library/publications/cia-maps-publications/index.html NASA Earth Observatory (publicdomain): http://earthobservatory.nasa.gov/

Reply: Figure 1 is original; no permits were required.

5. Please review your reference list to ensure that it is complete and correct. If you have cited papers that have been retracted, please include the rationale for doing so in the manuscript text or remove these references and replace them with relevant current references. Any changes to the reference list should be mentioned in the rebuttal letter that accompanies your revised manuscript. If you need to cite a retracted article, indicate the article’s retracted status in the References list and also include a citation and full reference for the retraction notice.

Reply: No retracted reference is included in the manuscript.

Additional Editor Comments: The manuscript needs a revision as recommended by the reviewers

Reply: We revised as the reviewer suggested.

Reviewers' comments: Reviewer's Responses to Questions

Comments to the Author

1. Is the manuscript technically sound, and do the data support the conclusions?

Reviewer #1: Partly

2. Has the statistical analysis been performed appropriately and rigorously?

Reviewer #1: N/A

3. Have the authors made all data underlying the findings in their manuscript fully available?

Reviewer #1: Yes

4. Is the manuscript presented in an intelligible fashion and written in standard English?

Reviewer #1: Yes

5. Review Comments to the Author

Reviewer #1: The manuscript by Wanchai Maleewong et al titled “Bacterial community structure of Anopheles hyrcanus group, Anopheles nivipes, Anopheles philippinensis, and Anopheles vagus from a malaria-endemic area in Thailand” is very important as it is focused on the investigation of the microbiota of some malaria vectors in Thailand. Despite the efforts put in place to control this deadly disease, it remains a public health mainly because of the resistance of mosquitoes to insecticides and the parasites themselves to drugs. Therefore, the use of microbiome based tools has been shown as an alternative vector control strategy. This work is then of great importance.

Reply: We would like to thank you for your kind suggestions and positive response. Your comments are encouraging and helpful.

However, some comments in order to improve the quality are as follows:

Abstract

In general, the authors need to add in the abstract the sample size and the software used for data analysis. Also, indicate if the DNA was extracted from the whole individual mosquito or the pool of mosquitoes, even the sequencing.

Reply: We modified as per your suggestion, “This study identified the bacterial microbiota in field-collected female adults of the Anopheles hyrcanus group and three Anopheles species, Anopheles nivipes, Anopheles philippinensis, and Anopheles vagus, from an endemic area in the southeastern part of Ubon Ratchathani Province, northeastern Thailand, near the Lao PDR-Cambodia-Thailand border. A total of 17 DNA libraries were generated from pooled female Anopheles abdomen samples (10 abdomens/sample). The mosquito microbiota was characterized through the analysis of DNA sequences from the V3-V4 regions of the 16S rRNA gene, and data were analyzed in QIIME2.” please see, page 2, lines 25-32.

In addition, the authors need to state the results of genetic diversity in terms of Alpha diversity and Beta diversity. Finally, it will be better to explain in one line the difference observed between these groups of bacteria.

Reply: We modified as per your suggestion, “Statistical difference in alpha diversity was observed between An. hyrcanus group and An. vagus and between An. nivipes and An. vagus, and beta diversity analyses showed that the bacterial community of An. vagus was the most dissimilar from other species.” please see, page 2, lines 34-36.

Line 31: “The results revealed the microbiota of four Anopheles species differed between pools of the same species and between species”. This result needs to be rephrased since Anopheles hyrcanus is stated here as a group and not a species.

Reply: We modified as per your suggestion, “A total of 3,442 bacterial ASVs were obtained, revealing differences in the microbiota both within the same species/group and between different species/group.” please see, page 2, lines 32-34.

Line 32: Before giving the predominant genus for each Anopheles groups, authors should give the predominant genus in the whole mosquito population of An. groups, even the total numbers of the identified bacterial. What about the predominant phylum? Also, the author should add the percentage to recover the overall abundance of these bacterial.

Reply: We modified as per your suggestion, “The most abundant bacteria belonged to the Proteobacteria phylum (48%-75%) in which Pseudomonas, Serratia, and Pantoea were predominant genera among four Anopheles species/group. However, the most significantly abundant genus observed in each Anopheles species/group was as follows: Staphylococcus in the An. hyrcanus group, Pantoea in the An. nivipes, Rosenbergiella in An. philippinensis, and Pseudomonas in An. vagus. Particularly, Pseudomonas sp. was highly abundant in all Anopheles species except An. nivipes.” please see, page 2, lines 36-42.

Line 38: “The present study provides new knowledge on the microbiota of four potential malaria vectors as a starting step towards understanding ….”. In the discussion, it is stated that it is the first study done in this context the authors should rewrite this sentence in the abstract such that it clearly appears that it is the first study and not only new knowledge as they said.

Reply: We changed “new knowledge” to “first study” as per your suggestion. “The present study provides the first study on the microbiota of four potential malaria vectors as a starting step towards understanding the role of the microbiota on mosquito biology and ultimately the development of potential tools for malaria control. Please see, page 2, lines 42-45.

Methods

Line 99: The authors said, “It is very difficult to identify species within the An. hyrcanus group based on morphology alone”. What about molecular identification? The authors should check if there is no protocol for that.

Reply: Anopheles hyrcanus group was used based on morphology alone (reference no. 33; Mosquito Taxonomic Inventory. Anopheles classification. Available from: https://mosquito-taxonomic-inventory.myspecies.info/node/11358.), until now, no accurate molecular identification is reported.

Line 100: “For that reason, we used our specimens belonging to this species group as An. hyrcanus group”, this sentence is not comprehensible and cannot be linked to the previous. Rewrite it to let it make sense.

Reply: We changed to “It is very difficult to identify species within the An. hyrcanus group based on morphology alone [33]. Then, we termed our specimens belonging to An. hyrcanus group. Please see, page 5, lines 104-106.

Line 102: The authors said, “Female Anopheles mosquitoes were dissected for detection and identification of Plasmodium parasites”. The parts dissected should be cited, are they abdomens, Head and thorax or ovaries…? Also, the method used to identify Plasmodium needs to state.

Reply: We modified to “The head and thorax were separated from the abdomen using sterile forceps for detection of Plasmodium parasites by molecular method [34].” Please see, page 5, lines 106-107.

Line 104: “Ten abdomens of each Anopheles species collected on the same day at around the same time were pooled …. bacterial community structure”. How many abdomens one pool? How many pools have been made for each species? These must be clear in the sentence. Also, add the estimated days collections.

Reply: We modified as per your suggestion, “Ten abdomens of each Anopheles species/group collected on the same day at around the same time were pooled per sample; An. hyrcanus group (HYR; n = 4 pools), An. nivipes (NIV; n = 5 pools), An. philippinensis (PHI; n = 4 pools), and An. vagus (VAG; n = 4 pools) and kept at -20°C until DNA extraction for further investigation of malaria and bacterial community structure.” Please see, pages 5-6, lines 109-113.

Line 115: “The presence of Plasmodium spp. parasites was investigated based on partial cox1 gene amplification”. The authors should well state the part of the mosquito used here for Plasmodium detection.

Reply: We modified to “The presence of Plasmodium spp. DNA was investigated based on partial cox1 gene amplification using DNA samples extracted from separated abdomen specimens.” Please see, page 6, lines 122-123.

Line 126: “From laboratory Plasmodium falciparum and Plasmodium vivax in-vitro cultures was used as 127 positive controls”: Why not used Plasmodium spp. as a positive control?

Reply: Now, no other laboratory Plasmodium spp. were available in our systems. However, COX-IF (5’ -AGA ACG AAC GCT TTT AAC GCC TG-3’and COX-IR (3’-ACT TAA TGG TGG ATA TAA AGT CCA TCC wGT-5’) primers are the conserve primers that can amplified other Plasmodium spp (Plasmodium falciparum, Plasmodium vivax, Plasmodium knowlesi, Plasmodium malariae, Plasmodium ovale wallikeri and Plasmodium ovale curtisi). Reference no, 34 (Echeverry et al. 2017).

Line 130: “The universal region-specific primers 341F (5’-CCT AYG GGR BGC ASC AG-3’) and 806R (5’-GGA CTA CNN GGG TAT CTA AT-3’…V3-V4 regions of the 16S rRNA gene were used.” Reference needs to be added here.

Reply: We added as per your suggestion, “The universal region-specific primers 341F (5’-CCT AYG GGR BGC ASC AG-3’) and 806R (5’-GGA CTA CNN GGG TAT CTA AT-3’) (NovogeneAIT Genomics Pte. Ltd., Singapore) tagged with sample-identifying barcodes to amplify the V3-V4 regions of the 16S rRNA gene were used.” Please see, page 7, lines 139-142.

Line 129: “Bacterial 16S rRNA gene amplification and sequencing”. In this paragraph, the authors should clearly give the details on the numbers of samples used, which parts have been used, which Anopheles species and how many pools?

Reply: We modified to “Seventeen DNA samples from pools of ten abdomens of each Anopheles species: An. hyrcanus group (HYR; n = 4 pools), An. nivipes (NIV; n = 5 pools), An. philippinensis (PHI; n = 4 pools), and An. vagus (VAG; n = 4 pools) were used for amplification and sequencing.” Please see, page 7, lines 137-139.

Line 134-138: The authors need to describe in detail the methodology for sample sequencing. Some previous studies have more describe this part. Also indicated whether the negative and positive controls have been used during the sequencing.

Reply: In the Illumina sequencing platform, quality control is carried out at each step of the process from DNA samples to the results, including sample testing, PCR, purification, library construction, and sequencing (no requirement for negative or positive control). We have added more information about quality control in the method: “At each step of the process, quality control was carried out to maintain the accuracy and reliability of the sequencing data according to the Illumina sequencing system (NovogeneAIT Genomics).” Please see, page 7, lines 145-147.

Results

In general, the authors need to indicate which season the mosquitoes have been collected, because some previous study show that the season could have a great impact on microbiota diversity.

Reply: We added sentence “covering three seasons (hot, rainy, and cold)” as per your suggestion, “The mosquitoes were collected throughout the study period, covering three seasons (hot, rainy, and cold).” Please see materials and methods section, page 5, lines 93-95.

Line 163: Before talking about the results on the bacterial composition, authors should first give the results of mosquitoes identification. It has been stated in the methodology that the four species used were the more predominant, what about the about? And which are the frequencies?

Reply: Results of mosquito identification and species composition were added as suggested, “A total of 2,659 Anopheles females belonging to one group (An. hyrcanus group) in the subgenus Anopheles and nine species/group (An. dirus s.l., An. karwari, An. kochi, An. maculatus group, An. minimus s.l., An. nivipes, An. philippinensis, An. tessellatus and An. vagus) in the subgenus Cellia were collected throughout study period. The four most abundant taxa were An. hyrcanus group (78.15%) followed by An. philippinensis (14.70%), An. nivipes (5.00%) and An. vagus (0.90%).” Please see, pages 8-9, lines 178-183.

Line 164. In this paragraph, the authors should mention the quality control statistic: the number of raw reads obtained, the number obtained after filtration, the rarefication…

Reply: We were represented in S1 Table, please see S1 Table. For clearer, we added “The output statistics are shown in S1 Table.” Please see, page 9, lines 186-187.

Line 180: “The microbiota of four Anopheles species differed between pools of the same species and between species”. Here the authors mentioned a difference between the pools of the same species, whereas the title on line 179 only mentions the differences between species. Moreover, in the following sentences, the differences they show are only those between species. What are about the differences between the same species?

Reply: Thank for your clarification, for better understanding, we modified and added more data to the S5 Table, and the added sentences “Shared ASVs were dominant bacteria and a small number of ASVs were different between samples in the same Anopheles species/group (S5 Table).” and “The microbiota of four Anopheles species/group were differed between the same and between species/group (S4 Table and S5 Table).” Please see S4 Table and S5 Table and revised text, page 9 , lines 193-194 and lines 200-201.

Line 187: “The Pseudomonadaceae was dominant in An. vagas and An. hyrcanus group, whereas …Carnobacteriaceae and Burkholderiaceae were abundant in An. hyrcanus group and An. vagas, respectively (Fig 3B). This paragraph should be complete, the authors mentioned the predominance in the 4 groups of mosquitoes but for the abundance, they just referred to An. vagas and An. hyrcanus, what about the 2 others? Also, to what the predominance and abundance here referred?

Reply: We have added already, please see the sentences “The Pseudomonadaceae was dominant in An. vagas and An. hyrcanus group, whereas Erwiniaceae was dominant in An. philippinensis and An. nivipes, followed by Moraxellaceae. Carnobacteriaceae and Burkholderiaceae were abundant in An. hyrcanus group and An. vagas, respectively (Fig 3B).” Please see, page 10, lines 207-210.

Line 205: “Elizabethkingia meningoseptica, Asaia krungthepensis, …species except An. nivipes”. Is this statement, are the authors sure that Asaia that they found where specifically krungthepensis? If not, they should better write Asaia sp.

Reply: Yes, we did discover the Asaia krungthepensis species that AVS named "OTU 13". Please see S2 Table and Fig. 4B. We also double-checked using BLASTn, and sequences were found corresponding to the species of Asaia krungthepensis (100% similarity).

Discussion

Line 238: “However, information on the gut microbiota of these four Anopheles species, which was characterized for further study when positive malaria samples are available for comparison”. This sentence must be rewritten; something is missing.

Reply: We modified as per your suggestion. “However, the limit information on the gut microbiota of these four Anopheles species/group were obtained, which was required for further study when positive malaria samples are available for comparison. Please see, page 12, lines 259-261.

Line 285: “The genus Rosenbergiella was the most abundant taxon in An. philippinensis”. It is just during the discussion that information is clear while it in the results, it is just noticed that this genus is among the most abundant. Since the idea is discussed, this part should be rephrased in the result such the reader is not surprised at the part discussion.

Reply: We modified as per your suggestion in results section. “For An. philippinensis, Rosenbergiella was the highest significant, followed by Bacillus, Acinetobacter, and Romboutsia.” Please see, page 10, lines 218-220.

Line 294: “Significantly, Pseudomonas aeruginosa and Carnobacterium maltaromaticum were found to be dominant in the guts of female An. hyrcanus group”. The authors should continue to use the term abdomen rather than gut, there is a slide difference.

Reply: We changed “guts” to “abdomens” as per your suggestion. “Significantly, Pseudomonas aeruginosa and Carnobacterium maltaromaticum were found to be dominant in the abdomens of female An. hyrcanus group.” Please see, page 14, lines 315-316.

Finally, we appreciate the reviewer very much for your kind suggestions, your comments are supportive and helpful.

---

## [Decision Letter · Decision Letter 1]

4 Jul 2023

PONE-D-23-12986R1Bacterial community structure of Anopheles hyrcanus group, Anopheles nivipes, Anopheles philippinensis, and Anopheles vagus from a malaria-endemic area in ThailandPLOS ONE

Dear Dr. Wanchai,

Thank you for submitting your manuscript to PLOS ONE. After careful consideration, we feel that it has merit but does not fully meet PLOS ONE’s publication criteria as it currently stands. Therefore, we invite you to submit a revised version of the manuscript that addresses the points raised during the review process.

ACADEMIC EDITOR: The manuscript needs revisions according to the reviewers opinion==============================

We look forward to receiving your revised manuscript.

Kind regards,

Shawky M Aboelhadid, PhD

Academic Editor

PLOS ONE

Reviewers' comments:

Reviewer's Responses to Questions

**Comments to the Author**

1. If the authors have adequately addressed your comments raised in a previous round of review and you feel that this manuscript is now acceptable for publication, you may indicate that here to bypass the “Comments to the Author” section, enter your conflict of interest statement in the “Confidential to Editor” section, and submit your "Accept" recommendation.

Reviewer #1: (No Response)

2. Is the manuscript technically sound, and do the data support the conclusions?

Reviewer #1: Yes

3. Has the statistical analysis been performed appropriately and rigorously? 

Reviewer #1: No

4. Have the authors made all data underlying the findings in their manuscript fully available?

Reviewer #1: Yes

5. Is the manuscript presented in an intelligible fashion and written in standard English?

Reviewer #1: Yes

6. Review Comments to the Author

Reviewer #1: The authors need to more describe in detail the bioinformatics analysis. Some previous studies have more describe this part. To establish whether alpha diversity differs across mosquito species for all samples, the authors need to consider the rarification depth (add this figure in supplementary data) of ASVs per sample, which will be sufficient to capture the typical low microbiota diversity in individual mosquitoes. You can see the link of the paper published recently: https://www.biorxiv.org/content/10.1101/2023.06.23.546313v1.

7. PLOS authors have the option to publish the peer review history of their article (what does this mean?). If published, this will include your full peer review and any attached files.

Reviewer #1: No

---

## [Author Response · Author response to Decision Letter 1]

16 Jul 2023

Point by point response to Editor and Reviewer comments.

ACADEMIC EDITOR: The manuscript needs revisions according to the reviewers opinion.

Reply: we modified as the reviewer comment.

Reviewers' comments: Reviewer's Responses to Questions

Comments to the Author

6. Review Comments to the Author

Reviewer #1: The authors need to more describe in detail the bioinformatics analysis. Some previous studies have more describe this part. To establish whether alpha diversity differs across mosquito species for all samples, the authors need to consider the rarification depth (add this figure in supplementary data) of ASVs per sample, which will be sufficient to capture the typical low microbiota diversity in individual mosquitoes. You can see the link of the paper published recently: https://www.biorxiv.org/content/10.1101/2023.06.23.546313v1.

Reply: We added more data to the S1 Figure and added the sentences, “Moreover, the abundance of ASVs was used to generate a rarefaction curve for estimating the species/group richness and diversity in the microbiota of four Anopheles species/group.” and “The alpha rarefaction curve represents the sequence to a sufficient depth of the sample, with most reaching saturation, which was sufficient to capture the scope of microbial diversity and indirectly reflect the abundances of our Anopheles species/group (S1 Fig).” as per your suggestion, Please see S1 Figure and revised text, page 8, lines 168–170, page 9, lines 191–194, and page 25, lines 557–562.

The S1 Figure did not include ACE and Simpson indices. For better understanding, the ACE index means the measuring of bacterial community richness can be also represented in the observed species and Chao1 indices. While Simpson index means bacterial diversity can also be represented in the Shannon index.

Finally, we would like to thank the Academic Editor and Reviewer for the supportive and helpful comments. We hope our revised manuscript would give you satisfaction.

---

## [Decision Letter · Decision Letter 2]

26 Jul 2023

Bacterial community structure of Anopheles hyrcanus group, Anopheles nivipes, Anopheles philippinensis, and Anopheles vagus from a malaria-endemic area in Thailand

PONE-D-23-12986R2

Dear Dr. Wanchai Maleewong, 

We’re pleased to inform you that your manuscript has been judged scientifically suitable for publication and will be formally accepted for publication once it meets all outstanding technical requirements.

Kind regards,

Shawky M Aboelhadid, PhD

Academic Editor

PLOS ONE

Additional Editor Comments (optional):

Reviewers' comments:

Reviewer's Responses to Questions

**Comments to the Author**

1. If the authors have adequately addressed your comments raised in a previous round of review and you feel that this manuscript is now acceptable for publication, you may indicate that here to bypass the “Comments to the Author” section, enter your conflict of interest statement in the “Confidential to Editor” section, and submit your "Accept" recommendation.

Reviewer #1: (No Response)

2. Is the manuscript technically sound, and do the data support the conclusions?

Reviewer #1: (No Response)

3. Has the statistical analysis been performed appropriately and rigorously? 

Reviewer #1: (No Response)

4. Have the authors made all data underlying the findings in their manuscript fully available?

Reviewer #1: (No Response)

5. Is the manuscript presented in an intelligible fashion and written in standard English?

Reviewer #1: (No Response)

6. Review Comments to the Author

Reviewer #1: (No Response)

7. PLOS authors have the option to publish the peer review history of their article (what does this mean?). If published, this will include your full peer review and any attached files.

Reviewer #1: No

<quillbot-extension-portal></quillbot-extension-portal>

---

## [Editor Report · Acceptance letter]

8 Aug 2023

PONE-D-23-12986R2 

Bacterial community structure of *Anopheles hyrcanus* group, *Anopheles nivipes, Anopheles philippinensis*, and *Anopheles vagus* from a malaria-endemic area in Thailand 

Dear Dr. Maleewong:

I'm pleased to inform you that your manuscript has been deemed suitable for publication in PLOS ONE. Congratulations! Your manuscript is now with our production department. 

Kind regards, 

on behalf of

Professor Shawky M Aboelhadid 

Academic Editor

PLOS ONE